# Estimation of COVID-19 Epidemiology Curve of the United States Using Genetic Programming Algorithm

**DOI:** 10.3390/ijerph18030959

**Published:** 2021-01-22

**Authors:** Nikola Anđelić, Sandi Baressi Šegota, Ivan Lorencin, Zdravko Jurilj, Tijana Šušteršič, Anđela Blagojević, Alen Protić, Tomislav Ćabov, Nenad Filipović, Zlatan Car

**Affiliations:** 1Faculty of Engineering, University of Rijeka, Vukovarska 58, 51000 Rijeka, Croatia; nandelic@riteh.hr (N.A.); ilorencin@riteh.hr (I.L.); car@riteh.hr (Z.C.); 2Clinical Hospital Centre, Rijeka, Krešimirova ul. 42, 51000 Rijeka, Croatia; zdravko.jurilj@gmail.com (Z.J.); alen.protic@uniri.hr (A.P.); 3Faculty of Engineering, University of Kragujevac, Sestre Janjić, 34000 Kragujevac, Serbia; tijanas@kg.ac.rs (T.Š.); andjela.blagojevic@kg.ac.rs (A.B.); fica@kg.ac.rs (N.F.); 4Bioengineering Research and Development Centre (BioIRC), Prvoslava Stojanovića 6, 34000 Kragujevac, Serbia; 5Faculty of Medicine, University of Rijeka, Ul. Braće Branchetta 20/1, 51000, Rijeka, Croatia; 6Faculty of Dental Medicine, University of Rijeka, Kresimirova 40/42, 51000 Rijeka, Croatia; tomislav.cabov@medri.uniri.hr

**Keywords:** artificial intelligence, COVID-19, epidemiology curve, genetic programming algorithm, regression modeling

## Abstract

Estimation of the epidemiology curve for the COVID-19 pandemic can be a very computationally challenging task. Thus far, there have been some implementations of artificial intelligence (AI) methods applied to develop epidemiology curve for a specific country. However, most applied AI methods generated models that are almost impossible to translate into a mathematical equation. In this paper, the AI method called genetic programming (GP) algorithm is utilized to develop a symbolic expression (mathematical equation) which can be used for the estimation of the epidemiology curve for the entire U.S. with high accuracy. The GP algorithm is utilized on the publicly available dataset that contains the number of confirmed, deceased and recovered patients for each U.S. state to obtain the symbolic expression for the estimation of the number of the aforementioned patient groups. The dataset consists of the latitude and longitude of the central location for each state and the number of patients in each of the goal groups for each day in the period of 22 January 2020–3 December 2020. The obtained symbolic expressions for each state are summed up to obtain symbolic expressions for estimation of each of the patient groups (confirmed, deceased and recovered). These symbolic expressions are combined to obtain the symbolic expression for the estimation of the epidemiology curve for the entire U.S. The obtained symbolic expressions for the estimation of the number of confirmed, deceased and recovered patients for each state achieved $R^{2}$ score in the ranges 0.9406–0.9992, 0.9404–0.9998 and 0.9797–0.99955, respectively. These equations are summed up to formulate symbolic expressions for the estimation of the number of confirmed, deceased and recovered patients for the entire U.S. with achieved $R^{2}$ score of 0.9992, 0.9997 and 0.9996, respectively. Using these symbolic expressions, the equation for the estimation of the epidemiology curve for the entire U.S. is formulated which achieved $R^{2}$ score of 0.9933. Investigation showed that GP algorithm can produce symbolic expressions for the estimation of the number of confirmed, recovered and deceased patients as well as the epidemiology curve not only for the states but for the entire U.S. with very high accuracy.

## 1. Introduction

According to 202 [1], the Coronavirus disease 2019 (COVID-19) is a respiratory and vascular disease that is caused by the severe acute respiratory syndrome coronavirus 2 (SARS-CoV-2). The outbreak of COVID-19 can be traced back to December 2019 in Wuhan province (China), although Apolone et al. [2] stated that the virus may have been actively spreading much earlier in Italy. The main transmission of COVID-19 from an infected to an uninfected person is through coughing, sneezing, singing, talking or breathing. The new infection [3,4] occurs when virus-containing particles (respiratory droplets or aerosols) get into the mouth, nose or eyes of uninfected people who are in the vicinity of the infected person.

The COVID-19 symptoms are variable, but generally include fever and cough [5,6,7]. However, people infected with COVID-19 may have different symptoms, and these symptoms may change over time. ISome patients have a high fever, cough and fatigue while in others have a low fever at the beginning and develop difficulty breathing weeks later. The symptoms of COVID-19 [6,8] may be non-specific such as fever and dry cough. The symptoms of COVID-19 can manifest 1–14 days after exposure to the virus [9]. The standard method for testing on COVID-19 is real-time reverse transcription-polymerase chain reaction (rRT-PCR) [10,11]. The rRT-PCR test is typically done using respiratory samples which are obtained using a nasopharyngeal swab. However, in some patients, a nasal swab or sputum sample may also be used.

Since the outbreak began, researchers from various fields have been extensively investigating this disease. Today, there are numerous research studies in which artificial intelligence (AI) was applied for the development of an epidemiological model of COVID-19. Most methods utilize machine learning (ML) algorithms on collected datasets. Most research focused on the application of neural networks, either applying regression methods on the datasets or analyzing the collected data in terms of time series, but other machine learning methods have been applied as well.

Several machine learning approaches have been used to model COVID-19 spread. One of the earliest papers [12] in this field is the utilization of Multi-Layer Perceptron (MLP) on a publicly available dataset [13] to estimate the number of confirmed, deceased and recovered patients on a global scale. In [14], the authors investigated the impact of COVID-19 on the financial movement of Crude Oil price and three U.S. stock indexes: DJI, S&P 500 and NASDAQ Composite. In this investigation, the system consists of the stationary wavelet transform (SWT) and bidirectional long short-term memory (BDLSTM) networks to predict the commodity and stock prices. In [15], the authors developed a modified stacked auto-encoder for modeling the transmission dynamics of COVID-19 epidemics in China. The data for this investigation were collected from 11th January 2020 to 27th February 2020, from WHO. Using this model, the authors performed forecasting of cumulative confirmed patients of COVID-19 across China from 20th January 2020 to 20th April 2020. Using the multiple-step forecasting, the estimated average errors of 6–10-day step forecasting were in the range from 0.73% to 2.27%. The combination of multiple machine learning approaches, including autoregressive integrated moving average (ARMA), cubist regression (CUBIST), random forest (RF), ridge regression (RIDGE), support vector regression (SVR) and stacking-ensemble learning, have been used [16] for the task of time series forecasting of one, three and six days ahead in ten Brazilian states with a high daily incidence. The results show that the models can generate accurate forecasting, achieving errors in a range of 0.87–3.51%, 1.02–5.62% and 0.95–6.90% for one, three, and six days ahead, respectively. The XGBoost classifier has been used [17] on 485 blood samples from infected patients in the region of Wuhan, China, to identify crucial predictive biomarkers of disease mortality. The utilized method predicted the mortality of individual patients more than 10 days in advance with an accuracy of 90%.

Deep Learning with LSTM network is used in [18] on the publicly available dataset provided by John Hopkins University and the Canadian health authority to forecast the COVID-19 outbreak in Canada. The results of the conducted investigation predict the possible ending point of this outbreak around June 2020. The hybrid Wavelet-autoregressive integrated moving average model and regression tree are used in [19] to forecast the number of daily confirmed patients for Canada, France, India, South Korea and the UK.

In this paper, the AI method, GP algorithm, is utilized, since this algorithm offers a possibility of creating mathematical expression from the given data which provides the best correlation between input and output data. Over the years, GP has been implemented in various fields such as curve fitting, data modeling and symbolic regression [20,21,22,23]; image and signal processing [24,25,26,27]; financial trading, time series prediction and economic modeling [28,29,30,31]; and industrial process control [32,33,34,35]. However, GP has also been implemented in medicine-based tasks. In [36], the authors applied GP to oral cancer prognosis. The dataset used in GP contained only 31 patients with feature selection of smoking, drinking, tobacco chewing, histological differentiation of SCC and oncogene p63. In this analysis, the authors achieved average scores of 83.87% accuracy and AUC score of 0.8341 for the classification task. In [37], the authors used GP and ANN to compare the performance on six medical classification problems, and these are breast cancer (benign or malignant), diabetes (positive or negative), gene (intron–exon, exon–intron or no boundary in DNA sequence), heart (diameter of a heart vessel is reduced by more than 50% or not), horse (horse with colic will die, survive or must be killed) and thyroid (thyroid hyperfunction, hypofunction or normal function). The results show that GP performs comparably to ANNs in classification problems. The authors of [38] developed prediction models for confirmed patients (CC) and death patients (DC) across the three most affected states Maharashtra, Gujarat and Delhi as well as the whole of India based on GP. The results show that the proposed models are highly reliable for short time series prediction of COVID-19 patients in India.

As seen from the literature overview, the implementation of AI is usually based on ANN and the GP algorithm implementation is seen in traces. The benefit of the GP algorithm over ANN is that after the training the symbolic expression is obtained, which can be used and manipulated in further analysis. The ANN in general will provide a trained architecture that is almost impossible to transform into mathematical expression due to the large number of interconnected neurons.

The global trend of COVID-19 has been on the exponential rise, with more outbreaks happening through time. This research focuses on the U.S. for two reasons. The first reason is the quality and quantity of the data available, which are highly precise in comparison to the data availability of other countries. As mentioned further in the paper, U.S. data are collected at many levels and the high number of tests performed allows data to be highly precise. This is true for the entire period of the data collection. The second reason is the extremely high number of cases exhibited in the U.S., with both recovery and death rates being relatively high, allowing for enough data to model these separate goals. After research of COVID-19, investigation of the application of various AI methods in COVID-19 spread and the literature overview of the GP algorithm, the following questions arise:
Is it possible to utilize a GP algorithm to obtain the symbolic expression for each U.S. state based on latitude and longitude of the central location of that state and the number of days since the outbreak began for the estimation of the number confirmed, deceased and recovered patients with high accuracy?Based on the obtained symbolic expressions for the estimation of the number of confirmed, deceased and recovered patients for each U.S. state, is it possible to formulate the symbolic expressions for the estimation of the number of confirmed, deceased and recovered patients for the entire U.S. with high accuracy?Is it possible to utilize three symbolic expressions for the estimation of the number confirmed, deceased and recovered patients for the entire U.S. to formulate symbolic expression for estimation of the epidemiology curve for the entire U.S. with high accuracy?

## 2. Materials and Methods

In this section, the publicly available dataset which was used in this paper is described. Then, the GP algorithm used to obtain the symbolic expressions for confirmed, deceased and recovered patients of each U.S. state is described. Based on these symbolic expressions, the methodology of determining the epidemiology curve for the entire U.S. is explained.

### 2.1. Dataset Description

As mentioned above, the publicly available dataset [13] was used to obtain symbolic expressions for confirmed, deceased and recovered patients using the GP algorithm. This dataset contains the number of confirmed, deceased and recovered patients for certain locations, for each day since the COVID-19 outbreak started. The locations in this dataset are defined with two parameters: latitude and longitude. Each state is defined with one location in terms of latitude and longitude. In total, 50 states were considered in these analyses. The federal district District of Columbia as well as inhabited territories such as American Samoa, Guam, Northern Marian Islands, Puerto Rico and the U.S. Virgin Islands were omitted from this investigation due to lack of data. In this study, the period used was from 22 January 2020 to 3 December 2020. The dataset is divided into three groups: confirmed, deceased and recovered patients. The dataset for confirmed/deceased/recovered patients for each state consists of the central location (latitude and longitude) of the state and the number of patients for each day since the outbreak started. The geographical locations and number of confirmed/deceased/recovered patients for each state are shown in Figure 1.

As shown in Figure 1, the blue dots indicate latitude and longitude locations that were used for each state as part of input data to obtain the symbolic expression for estimation of confirmed/deceased/recovered patients. In Figure 1, it can be noticed that the highest numbers of confirmed patients are in Texas, California and Florida followed by Illinois and New York. In Figure 1, it can be noticed that the highest numbers of deceased patients are in North Carolina, Utah, Colorado, Georgia and New Mexico. In Figure 1, it can also be noticed that the highest numbers of recovered patients are in states that have the highest number of confirmed patients, namely Texas, California and Florida.

As mentioned above, the publicly available dataset was used to obtain symbolic expressions for confirmed, deceased and recovered patients using the GP algorithm. The initial form of the dataset was in time-series form starting from 22 January 2020 to 3 December 2020. For each state, the central location of that state is given (latitude and longitude) as well as the number of confirmed, deceased and recovered patients in the above-mentioned period. It should be noted that, in this investigation, only U.S. states were considered while the federal district (District of Columbia), as well as inhabited territories (American Samoa, Guam, Northern Marian Islands, Puerto Rico and the U.S. Virgin Islands) were omitted. The reason they were excluded is that only the states were considered and that for previously mentioned territories the number of confirmed/deceased/recovered patients is missing for some dates. The data for each state were transformed from the time-series form into regression form. Thus, for the central location of each state, there are only 317 instances used to train and test symbolic expression obtained by the GP algorithm. Each of the 317 instances of the regression data consisted of three input variables, namely latitude and longitude of the state central location and the day from which the outbreak started, while the output variable was the number of confirmed/deceased/recovered patents for a specific date. The last ten instances of the confirmed number of patients dataset for the state of Alabama are shown in Table 1.

As shown in Table 1, the latitude and longitude of the central location for Alabama that represents the first two input variables are constant while the only changing input variable is the number of days since the outbreak began. The output variable in this case represents the number of confirmed patients for each day since the outbreak began. For each state, these 317 instances were shuffled to prevent overtraining of the GP algorithm and divided into training and testing datasets in the ratio of 80:20. The training dataset consisted of 254 instances and was used in GP to obtain symbolic expression, while the remaining 63 instances were used to test the symbolic expression and measure the $R^{2}$ score. For each state, the data from the modified dataset was divided into training and testing data in a ratio of 80:20. This means that 80% of the dataset or 254 instances were used to obtain symbolic expression using the GP algorithm while the remaining 20% or 63 instances were used to test the obtained symbolic expression and calculate the $R^{2}$ value. In each GP execution, the entire dataset for the state was randomly shuffled and then divided into training and testing datasets to prevent overfitting.

In the case of confirmed/deceased/recovered patients for each state, the data were first randomly shuffled and then divided into a ratio of 80:20. This means that the 80% dataset for confirmed/deceased/recovered patients of each state was used to obtain the symbolic expression for the estimation of the number of confirmed/deceased/recovered patients. The estimation accuracy in terms of the $R^{2}$ score of the obtained symbolic expression was then evaluated on 20% of the dataset and 80% of the training dataset for each state. If the $R^{2}$ number was below 0.99 in both the training and testing dataset, the execution was repeated by shuffling the dataset and then dividing it in the same ratio. The procedure was repeated 30 times, and, if the $R^{2}$ score in both training and testing was below 0.99, the GP algorithm was executed for another state (using the dataset for the next state). After all symbolic expressions were obtained for the estimation of the number of confirmed/deceased/recovered cases, the analysis of the obtained symbolic expressions was performed. For those symbolic expressions that did not achieve the 0.99 $R^{2}$ score, the highest achievable $R^{2}$ score was chosen. To put those percentages into better perspective, for each state, there are 317 instances in terms of the number of confirmed/deceased/recovered patients for each day since the outbreak began. As stated, 80% of the dataset for each state was used for training, i.e. 254 instances, while the remaining 20% (63 instances) were used to test the obtained symbolic expression or in other words to calculate the $R^{2}$ score. By doing such a rigorous procedure, overfitting was avoided.

The reason these input attributes were selected is that these values are not entirely relevant to the spreading of COVID-19, but a question arises if these three input variables (latitude, longitude and day) could be used to obtain the symbolic expressions for the estimation of the number of confirmed/deceased/recovered patients for each state. The other reason these variables were chosen was because these input values were already provided in the dataset. The reason the authors chose the U.S. for this investigation is that the number of confirmed/deceased/recovered patients is well documented for each state. While the inclusion of other variables such as population density, climate, travel and migration would contribute to the process of obtaining symbolic expressions, we wanted to investigate if it is possible to obtain symbolic expressions for the estimation of the number of confirmed, deceased and recovered cases for each state using only latitude, longitude and the number of days since the outbreak began.

### 2.2. Genetic Programming Algorithm

Genetic Programming (GP) algorithm can be described as the combination of machine learning (ML) methods and Evolutionary Algorithms (EA). As in the case of most supervised ML models, the dataset must be divided into training and testing portions. The training dataset is used in GP to obtain symbolic expression, which correlates input values with the output. The testing dataset is used to test the obtained symbolic expressions. This portion of the dataset is unseen by the symbolic expression and GP for that matter.

The benefit of utilizing the GP in comparison to other machine learning methods is the shape of models generated by it, formulated as equations that transform the set of inputs to the output goal. Because these equations utilize basic mathematical functions, they can, after only minor modifications, be utilized in any software supporting them. This is important in multi-discipline goals such as the one explored here because epidemiological or medical staff may not have the equipment and software necessary to utilize the models generated by a neural network. Even among the engineering staff, different versions of libraries and unfamiliarity with the programming language used in the model creation can cause issues during model interpretation. Simple, language-agnostic, mathematical equations can easily be shared and implemented in new or existing software.

The GP algorithm starts the execution by creating the initial population which is then propagated throughout the predefined number of generations. In each generation, the population members compete to become parents of the next generation usually using tournament selection. After the selection of the best population members using the aforementioned selection procedure, the genes between two or more population members are exchanged using crossover operation, or genes of population members are randomly selected and changed in mutation operation.

To obtain the best symbolic expressions for estimation of confirmed/deceased patients for each state using GP algorithm, the input–output variables and GP parameters must be defined. In the case of confirmed/deceased/recovered patients for each state, the input and output variable representation is shown in Table 2.

As shown in Table 2 the input variables in GP algorithm for development of symbolic expression for estimation of confirmed/deceased/recovered patients for each state are latitude ($X_{0}$), longitude ($X_{1}$) and the day ($X_{2}$) since the outbreak began. The output variable in symbolic expression for estimation of confirmed/deceased/recovered patients is the number of confirmed/deceased/recovered patients, respectively.

The initial set of solutions in the GP is named the initial population and is typically randomly generated. It can either be generated using the full-size tree (up to the maximal defined value), called a full method, or a grow method, which can stop before a maximal size is reached. The combination of the two, called ramped half-and-half, was used in the presented research. Nodes of the generated trees are randomly selected from the available ones in the geneset.

The continuity of the generated equations is guaranteed by the fact the GP uses modified versions of certain mathematical operations. For example, in the case of a square root $\sqrt{x}$, the operation is implemented as a square root of an absolute value ($\sqrt{\left|x\right|}$), to avoid calculating square roots of negative values. Another common discontinuity is division by zero. In the case of division by zero, or near-zero values, a protected division is used, which returns the value of 1.0 [39,40,41]. These modified operations vary between GP implementations and must be taken into account when the generated models are implemented.

#### Fitness Function

After population initialization and in each generation, the population members must be evaluated to determine how well they perform before performing the selection. This task is achieved with fitness measure which is a primary mechanism for giving a high-level statement of the problem’s requirements to the GP system.

The GP syntax trees are interpreted utilizing executing the nodes in the tree in a specific order that guarantees that nodes are not executed before the value of their arguments is known. This procedure is achieved by traversing the tree recursively starting from the root node, and executing the evaluation of each node after the values of its children are known. In this paper, the mean absolute error (MAE) is utilized as a fitness function for evaluation of population members, which can be written in the following form:
(1)$$MAE=\frac{\sum_{i=1}^{n}\left|y_{i}-x_{i}\right|}{n},$$
where $y_{i},x_{i}$ and *n* represent the prediction, true value and number of instances, respectively. After each GP algorithm execution, the symbolic expression is obtained on training portion of the dataset. This symbolic expression is then evaluated on the testing portion of the dataset with coefficient of determination ($R^{2}$). The $R^{2}$ metric of each symbolic expression is calculated using the mathematical equation which can be written in the following form
(2)$$R^{2}=1-\frac{S_{RESIDUAL}}{S_{TOTAL}}=1-\frac{\sum_{i=0}^{m}{\left(y_{i}-{\hat{y}}_{i}\right)}^{2}}{\sum_{i=0}^{m}{\left(y_{i}-\frac{1}{m}\sum_{i=0}^{m}y_{i}\right)}^{2}}.$$

The $R^{2}$ metric compares two set of solutions, and these are the real data *y* and the data obtained by the model $\hat{y}$. This means that the $R^{2}$ metric calculates the amount of variance contained inside the data *y*, which is explained by the data $\hat{y}$ as a model output. The result of $R^{2}$ metric is in range from 0 to 1 where 1 indicates that there is no variance between the real data and the data obtained by the model while 0 value indicates that none of the variance in the real data is explained by the model.

The improvement of solutions is achieved using evolutionary computation operations:
Crossover is taking two selected solutions and combining them into a new, children, solution (influenced by the crossover coefficient).Mutation is randomly modifying an existing solution from the previous and copying it to the current generation (influenced by the subtree, hoist and point mutation coefficients).Reproduction is copying the solutions from the previous to the current generation without modification (influenced by the maximal sample’s coefficient).

The solutions to be used in the above operations are determined using tournament selection, which is a fitness proportional selection. The process of selection and modification using EC operations is performed until the termination criteria are reached, termination criteria being either the selected number of generations being reached or the fitness value falling below a pre-specified threshold. The final hyperparameter of the GP which needs to be described is the parsimony coefficient. The tendency exists for the solutions to grow through the generations. Sometimes, this results in a better solution, but, in other cases, the growth generates larger equations without significant gains in the fitness function. The larger solutions are of higher computational complexity, slowing both the training process and the later use of the generated models. To combat this, the fitness function of the solutions may be lowered depending on their size. The amount of this is modified with the parsimony coefficient, the larger value of which penalizes the large solutions more.

The utilized values of hyperparameters are given in Table 3. The hyperparameters are randomly selected from the given ranges and training is performed. In the case the desired $R^{2}$ value is not reached once the GP training is complete, new hyperparameters are randomly selected and the process is repeated. In addition to hyperparameter values, GP uses a function set of mathematical functions to insert into symbolic expressions: addition, subtraction, multiplication, division, square root, maximum and minimum of values, absolute value and natural logarithm.

As the table shows, not all of the goals utilize the same ranges. Initial hyperparameter ranges for all goals were equal. During the research, certain hyperparameter values for certain goals were increased to obtain higher quality solutions, in the case that the smaller range did not provide the necessary performance. Initially, smaller ranges were preferred, due to shorter training times, but were increased if the models did not achieve the required error.

In this investigation the computer hardware used consisted of CPU Intel Core I5-4570 with a base clock of 3.20 GHz, 8 Gb of DDR3 RAM. The time required to obtain the symbolic expression for the estimation of the number of confirmed/deceased/recovered patients for each state was approximately 4/3/5 min, respectively. If in each GP algorithm execution, the obtained symbolic expression for the estimation of the number of confirmed/deceased/recovered patients achieved a high $R^{2}$ score, then the total number of GP algorithm executions would be 150, i.e., 150 symbolic expressions. The total (ideal) time to obtain these symbolic expressions would be 600 min (10 h). However, this is not the case since for some symbolic expressions for the estimation of the number of confirmed/deceased/recovered patients there were more than 10 GP algorithm executions required to obtain symbolic expressions with high accuracy. Thus, the approximate time to obtain all 150 symbolic expressions (for each state, three symbolic expressions for estimation of confirmed, deceased and recovered patients) took approximately two working days.

### 2.3. Epidemiology Curve

To define the epidemiology curve for a specific area first the symbolic expressions for the estimated number of confirmed, deceased and recovered patients must be obtained first. After the aforementioned symbolic expressions are obtained, the epidemiology curve can be calculated using the following expression:(3)$$y=y_{confirmed}-y_{deceased}-y_{recovered},$$
where $y_{confirmed}$ represents the total estimated number of confirmed patients, $y_{deceased}$ represents the total estimated number of deceased patients and $y_{recovered}$ represents the total estimated number of recovered patients. Thus, the estimated epidemiology trend for a specific country can be calculated as the difference among confirmed, deceased and recovered patients. In this case, the total estimated number of confirmed patients is calculated as the sum of all symbolic expressions obtained for 50 U.S. states and is written in the following form:(4)$$y_{confirmed}=\sum_{n=1}^{N}y_{Ci},\;i=1,\ldots ,50,$$
where $y_{Ci}$ represents the estimated number of confirmed patients for each state. The total estimated number of deceased patients for the entire U.S. is obtained in the same way using the following expression:(5)$$y_{deceased}=\sum_{n=1}^{N}y_{Di},\;i=1,\ldots ,50,$$
where $y_{Di}$ represents the estimated number of deceased patients for each states which is obtained using symbolic expression for the specific state. The same procedure is used to obtain total estimated number of recovered patients in the entire U.S. using the following mathematical expression:
(6)$$y_{recovered}=\sum_{n=1}^{N}y_{Ri},\;i=1,\ldots ,50,$$
where $y_{Ri}$ represents the estimated number of recovered patients for state *i*.

## 3. Results and Discussion

In this section, the results are presented and discussed. First, the symbolic expressions for the numbers of confirmed, deceased and recovered patients are presented, followed by the symbolic expression for the estimation of the epidemiology curve for the entire U.S.

### 3.1. Symbolic Expression for Estimation of the Number of Confirmed Patients for Each State and the Entire U.S.

The procedure of obtaining the symbolic expression of estimation of confirmed patients for each state was performed on the dataset [13], which was split into training and testing portions in ratio 80:20. This means that 80% of the dataset was used for training or in other words for obtaining the symbolic expressions, while 20% of the dataset was to obtain the $R^{2}$ score using Equation (Equation 2). In each iteration of the GP algorithm, the GP parameters were randomly selected from the pre-specified range given in Table 3. The majority of obtained symbolic expressions used for estimation of confirmed patients for each state are too large to be presented in this paper. Instead, examples of obtained symbolic expressions are shown for the estimation of confirmed patients for Maryland and Virginia, which achieved accuracy on the testing dataset of 0.99334 and 0.9915, respectively.
(7)$$\begin{array}{cc} y_{ConfMaryland}= & X_{0}X_{2}\sqrt{|X_{2}|}-\min(X_{0}X_{2}\sqrt{|X_{2}|},|X_{1}|(-\min(X_{2}^{2}-14547.7,X_{2}\left|X_{1}\right|,\\ \; & |X_{1}{|}^{2},\left(X_{2}^{2}-14547.7\right)\log(X_{2}|X_{2}|-14547.7))+X_{2}|X_{2}{\left|-14547.7\right)}^{\frac{1}{2}}) \end{array}$$
(8)$$\begin{array}{cc} y_{ConfVirginia}= & X_{2}\max(\left(X_{2}-\frac{X_{1}}{X_{0}}\right)\max(-59018.9,0.000021\max\left(1,X_{0}\right)\\ \; & \sqrt{\sqrt{X_{2}-\min\left(X_{0},X_{2}\right)}\max\left(\frac{66744.2}{X_{0}},\frac{X_{1}}{X_{0}},X_{2}\right)}),\min\left(X_{0},-\frac{X_{1}}{X_{0}}+X_{1}+X_{2}\right))\\ \; & (X_{2}-\min(X_{2},\max(\sqrt{X_{2}-\max\left(X_{2},\sqrt{\min(-59018.9,\frac{X_{1}}{X_{0}},X_{2},\sqrt{\min(X_{0},X_{2})})}\right)},\\ \; & \min(\max\left(\frac{X_{1}}{X_{0}},\frac{X_{1}}{\left(X_{2}-X_{0}\log\left(X_{2}\right)\right)\max\left(0.000042X_{0},\frac{X_{2}}{X_{1}}\right)}\right),\\ \; & \sqrt{X_{2}-\max\left(X_{2},\sqrt{\min(-59018.9,X_{2})}\right)}+X_{0}))){)}^{\frac{1}{2}} \end{array}$$

These two symbolic expressions of confirmed patients for Maryland and Virginia were chosen due to the simplicity of symbolic expressions. In these equations, $X_{0}$ represents latitude, $X_{1}$ represents longitude and $X_{2}$ is the days elapsed since the start of the dataset. The GP parameters used for obtaining the symbolic expressions of confirmed patients for each state is given in Table A1. The table of results is given in Appendix A (Table A1).

As shown in Table A1, the population tended towards the lower values in the range, with most of the solutions using population sizes smaller than 1000 in all cases. Most solutions used the number of generations higher than 150, up to a maximum of 194/200. For the tree depth, the values for all states models were around the middle of the available range. Crossover coefficient tended towards the lower side of the range, with some of the solutions even using the 0.9 minimum available value, the same being true for all mutation coefficients. Constant ranges are large, indicating the lack of information contained within the dataset inputs. Parsimony coefficients tend towards the higher side of the available range, indicating that the models ran into bloating issues that needed to be curbed. All $R^{2}$ scores achieved are above 0.9, the lowest being 0.94 for Vermont. The distribution of achieved results for each symbolic expression in terms of $R^{2}$ score for each state is shown in Figure 2a.

As shown in Figure 2a, the obtained symbolic expressions for estimation of confirmed patients for each state have very high accuracy. The highest accuracy (higher than 0.999) was achieved with the symbolic expressions for New York, North Dakota Oregon, South Carolina, Virginia, West Virginia and Wisconsin. To compare the accuracy of obtained symbolic expression for confirmed patients with the real data all symbolic expressions for estimation of confirmed patients for each state are summed up using Equation (Equation 4). The accuracy of summed symbolic expression for the estimation of confirmed patients for the entire U.S. is calculated using Equation (Equation 2). In Figure 2b, the summed symbolic expression for estimation of confirmed patients is compared to the real data for the entire U.S.

As indicated in Figure 2b, the accuracy of symbolic expression for the estimation of the number of confirmed patients achieved the $R^{2}$ score of 0.9992. This score is graphically validated in comparison to the real data. The increase in the number of confirmed cases in the observed period can be best described throughout political and social events that have occurred in the aforementioned period. In Figure 2b, it can be noticed that in the first 60 days since the outbreak started there is almost negligible growth in the number of confirmed patients when compared to the interval from 60 to 317 days. According to MD [42], the virus had been circulating undetected at least since January 2020, and possibly as early as November 2019. The first reported case of COVID-19 in the U.S. was reported on 21 January 2020. From 21 January to 23 February 2020, 14 cases of COVID-19 were reported in six states. However, the outbreak appeared contained through February 2020, although the CDC warned the American public for the first time on 25 February 2020 (35th day since the outbreak began) to prepare for a local outbreak [43]. In the last week of February, several large events contributed to the further spreading of COVID-19 in Louisiana, Massachusetts and Georgia [43]. On 12 March 2020, the number of confirmed cases in the U.S. exceeded 1000 [44]. According to Liptak [45], the White House on 16 March advised the general population to avoid gatherings of more than 10 people, and on 19 March the State Department [46] advised U.S. citizens to avoid all international travel. According to Khazan [47], by the middle of March, all 50 states were able to perform tests with a doctor’s approval but the number of available test kits remained limited, which means that the true number of people infected was much higher than reported. In this period, federal and state agencies began taking urgent steps to prepare for a surge of hospital patients, establishing additional places for patients in the case hospitals became overwhelmed and the manpower from the military and volunteer armies were called up to help construct the emergency facilities. Although the government in this period responded by preventing rallies and providing testing facilities and hospital capacities for a growing number of confirmed patients, the general population began to protest against government-imposed lockdowns. The first protest was in Michigan on 15 April 2020 (85th day since the outbreak began) where an estimated 3000 people took part in the protest [48]. Following the protest in Michigan, anti-lockdown protests were held in every state where protesters were in the range from 100 s to 1000 s. Additionally, there were 450 major protests (Black Live Matters) which were held in cities and towns across the U.S. due to racially charged events [49]. These protests that occurred in a period between 90 and 150 days since the outbreak began certainly had a major influence on the rapid virus spread as well as the growth in the number of confirmed patients. In the period between 100 and 250 days since the outbreak in the U.S. started, more than 150 health professionals sent a letter to the federal government in which they requested a lockdown of 6–8 weeks. They believed that this would restore the country by 1 October 2020 [50,51,52]. If the government enabled the lockdown, this could prevent rapid growth of the number of confirmed patients; however, the government’s negligence to the multiple demands of health professionals also contributed to the rapid spread of the virus. The additional contributing factor to the rapid spread of the virus is the motorcycle rally in South Dakota, which more than 400,000 people attended [53]. The massive gathering resulted in more than 300 confirmed patients from 20 states [53,54]. In the last period, from 250 to 317 days since the outbreak began, the presidential elections campaign contributed to the additional spreading of the virus. It is reported that the aftermath of presidential elections campaigning increased the number of confirmed patients by 35% [55]. At the end of the investigated period, all previously mentioned political and social events had some influence on the virus spread, which resulted in more than 14,000,000 confirmed patients by 317 days since the outbreak began.

### 3.2. Symbolic Expression for Estimation of the Number of Deceased Patients for Each State and the Entire U.S.

The procedure for obtaining a symbolic expression for the estimation of deceased patients for each state is the same as for the symbolic expressions obtained for the estimation of confirmed patients for each state. Most of the obtained expressions are too large to be presented in this paper. Instead, the examples of the two smallest symbolic expressions are shown for the estimation of the number of deceased patients for Hawaii and Idaho, which achieved accuracy on a testing dataset of 0.9905 and 0.9906, respectively.
(9)$$\begin{array}{cc} y_{deceasedHawaii}= & \max(-X_{0}+2X_{2}-354.196,\\ \; & \sqrt{\log\left(\sqrt{X_{2}-X_{0}}\right)\left(-\left(X_{2}-2X_{0}\right)\log\left(\sqrt{X_{2}-X_{0}}\right)-X_{0}+X_{2}\right)}). \end{array}$$
(10)$$\begin{array}{c} y_{deceasedIdaho}=\frac{0.0000132271X_{1}X_{2}^{2}\left(2X_{2}-\frac{X_{0}X_{1}}{X_{2}}\right)\min\left(\frac{X_{1}}{X_{0}},\frac{X_{2}}{X_{1}}\right)}{X_{0}\sqrt{X_{0}}}. \end{array}$$

Equations (Equation 9) and (Equation 10) represent two symbolic expressions for the estimation of the number of deceased patients for Hawaii and Idaho. $X_{0}$ and $X_{1}$ represent latitude and longitude of the central location of each state, respectively. $X_{2}$ represents a specific day calculated from the date at which the COVID-19 outbreak started (22 January 2020). In Table A2, the GP parameters used to obtain the best symbolic expression of deceased patients for each state are given with the achieved $R^{2}$ accuracy. Individual results are given in Table A2.

As was the case previously, the population ranges tended towards the higher possible value, but there are selected values all across the range. Many models converged with the number of generations near the lower end of available values. Most solutions used the higher end of the available values for the number of tournament entries. Many initial tree size values tended towards having six as the lower bound. Similar to the previously observed case, the values for crossover and mutation probabilities tend towards the lower possible range. The value of the maximum number of samples is shown to have been selected across the entirety of the range. The range of constant used is also large for the best-selected solutions. Parsimony coefficient values tend to be around the middle of the range for most cases. The achieved $R^{2}$ score values of obtained symbolic expression for deceased patients estimation for 50 states are in a range between 0.9404 and 0.9998. The lowest $R^{2}$ score value was achieved in the case of Washington while the largest $R^{2}$ score value was achieved in the case of Florida.

The achieved $R^{2}$ score for each state is shown in Figure 3a.

As shown in Figure 3a, all symbolic expressions for estimation of deceased patients obtained for each state are estimating the number of deceased patients with high accuracy. The symbolic expressions that achieved accuracy higher than 0.999 are for Arkansas, California, Florida, Illinois and Missouri. To compare the accuracy of obtained symbolic expressions for deceased patients with the real data, all symbolic expressions for each state are summed up using Equation (Equation 5). The accuracy of the summed symbolic expression for estimation of deceased patients for the entire U.S. is calculated using Equation (Equation 2). In Figure 3b, the summed symbolic expression for deceased patients estimation is compared to the real data for the entire U.S.

As shown in Figure 3b, the summed symbolic expression for the estimation of the number of deceased patients for the entire U.S. estimates the number of deceased patients with high accuracy when compared to the real data (data from the dataset). The achieved $R^{2}$ accuracy of symbolic expression for the estimation of the number of deceased patients for the entire U.S. is equal to 0.9997.

As shown in Figure 3b, the number of deceased patients in the first 50 days since the outbreak began is very small. The first recorded deceased patient in the U.S. was in California on 2 February on 2020 [56]. On 11 April 2020, the number of deceased patients in the U.S. became the highest in the world with the number of deceased patients reaching 20,000 [57]. The anti-lockdown and Black Lives Matter protests which occurred between April and July in various states had a huge contribution to the increase of confirmed and deceased patients. By 27 May, 100,000 patients had died from COVID-19 [58]. On 22 September, the number of deceased patients passed 200,000 [59]. The presidential election campaign also had some influence on the number of deceased patients since the number surpassed the value of 250,000. According to Woolf et al. [59], COVID-19 has become deadlier than heart disease and cancer. The mortality rates from COVID-19 poses the threat to different age groups. The investigation showed that COVID-19 had become the third leading cause of death for patients aged 45–84 years and the second leading cause of death for those aged 85 years or older. For example, between 1 October (254th day) and 3 December (317th day) the number of COVID-19 deaths tripled, from 826 to 2430 deceased per day. However, it should be mentioned that, on 21 April (91st day), at the height of the spring surge, the number of COVID-19 deaths was 2856. The record-breaking number of deceased patients, which is indicated in the last 63 days in Figure 3b, indicates that lethality may increase further as transmission increases with holiday travel and gatherings.

### 3.3. Symbolic Expression for Estimation of the Number of Recovered Patients for Each State and the Entire U.S.

The procedure of obtaining symbolic expressions for the estimation of the number of recovered patients for each state is the same as the procedure used to obtain the symbolic expression for the estimation of the number of the confirmed and deceased patients for each state. Most of the symbolic expressions used to estimate the number of recovered patients for a particular state are too large to be presented in this article. Instead, two simple symbolic expressions are given, which are obtained for estimation of the numbers of the recovered patients for Washington and Wyoming that achieved accuracies on the testing dataset of 0.9929 and 0.9966, respectively.
(11)$$y_{RecoveredWashington}=-\frac{0.0000508903X_{1}X_{2}^{2}\log(|X_{1}\left|\right)\min\left(X_{2},X_{2}-X_{0}\right)}{\min(\sqrt{\frac{X_{2}^{3/2}}{X_{0}}},X_{2}-X_{0})}.$$
(12)$$y_{RecoveredWyoming}=\frac{X_{2}\left(\frac{X_{2}\sqrt{|X_{1}|}}{\log\left(X_{2}\right)\left|\frac{2X_{2}-X_{1}}{|\sqrt{|X_{2}-X_{1}|}-X_{0}|}-X_{0}\right|}+X_{2}\right)}{|\frac{2X_{2}}{|\sqrt{|X_{2}-X_{1}|}-X_{0}|}-X_{0}|}.$$

As above, in Equations (Equation 11) and (Equation 12), $X_{0}$ and $X_{1}$ are the latitude and longitude of central state locations, respectively, while $X_{2}$ is the day since the COVID-19 outbreak started (22 January 2020). In Table A3, the symbolic expression for the entire U.S. is given with GP parameters used to obtain the symbolic expression, symbolic expression, and the $R^{2}$ accuracy. In Table A3, the GP parameters used to obtain the best symbolic expression for the estimation of the number of recovered patients for each state is given with the achieved $R^{2}$ score. As in previous cases, individual results may be found in the Appendix (Table A3).

The GP parameters shown in Table A3 used to obtain the symbolic expression for estimation of recovered patients in each state were randomly selected from the pre-specified ranges shown in Table A3.

Most solution populations were close to the middle of the available range. The same is true for the selected number of generations, while the tournament size tended towards the higher end of the available range. It can be noticed that the values of the operation coefficients, as in the previously observed cases, tend towards the lower end of the range. A number of the selected maximal samples tended towards the higher end of the range. The values of parsimony coefficients were equally distributed across the range. Values of constant ranges are large as they were in both initial cases.

As shown in Table A3, all symbolic expressions obtained for estimation of recovered patients for each state achieved very high accuracy, which was measured in terms of $R^{2}$ score. The $R^{2}$ score is in a range from 0.9797 to 0.99955. The lowest accuracy value achieved was in the case of Vermont, while the highest accuracy value was achieved in the case of Minnesota. In Figure 4a, the $R^{2}$ accuracy is shown on U.S. map.

As shown in Figure 4a and Table A3, it can be noticed that all symbolic expressions achieved very high accuracies. The symbolic expressions obtained for estimation of the number of the recovered patients that achieved $R^{2}$ accuracies higher than 0.999 are those symbolic expressions obtained for Idaho, Minnesota, Ohio, Tennessee and Texas. To compare the estimation of recovered patients for the entire U.S. with the real data, the symbolic expression of the entire U.S. was obtained as the sum of 50 symbolic expressions (each symbolic expression for one state) using Equation (Equation 6). The results obtained with the symbolic expression for the entire U.S. and the real data are used in Equation (Equation 2) to obtain the $R^{2}$ value to measure the accuracy of the symbolic expression for estimation of the number of recovered patients for the entire U.S. In Figure 4b, the comparison of the estimated number of recovered patients achieved with the obtained symbolic expression for the entire U.S. with the real data is shown.

As shown in Figure 4b, the number of recovered patients in the entire U.S. is exponentially growing. In the first 100 days since the outbreak began (22 January 2020), the number of recovered patients was very small when compared to the remaining 217 days. After 200 days, the number of recovered patients reached almost 2,500,000. It is interesting to notice that, in the last 20 days, the number of recovered patients is growing much faster than ever before, since the number of recovered patients in that 20 days is almost 1,000,000. The symbolic expression for the estimation of recovered patients for the entire U.S. estimates the number of recovered patients with high accuracy when compared to the real data (data from the dataset). Another indicator of how accurate is the symbolic expression is the $R^{2}$ value of 0.9996 calculated using Equation (Equation 2).

At the beginning of the virus outbreak in the U.S., the first 80 days the number of recovered patients has a similar trend as the number of confirmed and deceased patients. As the number of confirmed patients increased due to various large gatherings such as protests and other manifestations, the number of recovered patients also increased. However, it should be mentioned that recovery from the disease was different for different age groups. Woolf et al. [59] shown that the mortality rate is higher for age groups above 35 and that, ofr the age group higher than 85, the death of virus in the U.S. is the second cause of death. Younger age groups are more likely to recover from the disease than the older ones.

#### Symbolic Expression for Estimation of Epidemiology Curve for the Entire U.S.

At this point, all the required components for defining symbolic expression for epidemiology curve estimation are defined. The symbolic expression for the estimation of the number of confirmed patients for the entire U.S. is defined with the summation of 50 symbolic expressions. Each symbolic expression is the expression for the estimation of the number of confirmed patients for a specific U.S. state. The same procedure is applied to obtain the symbolic expression for the estimation of the number of deceased patients for the entire U.S. With these three equations, the epidemiology curve for the entire U.S. can be calculated using Equation (Equation 3). The obtained symbolic expression for epidemiology curve estimation is used with the real data to calculate the accuracy of the aforementioned symbolic expression in terms of $R^{2}$ metric using Equation (Equation 2). The achieved $R^{2}$ value of the symbolic expression for estimation of the epidemiology curve is equal to 0.9933. In Figure 5, the estimated epidemiology curve is compared with the real data.

As shown in Figure 5, the general trend of infected (active) patients is still rising. For the first 250 days since the outbreak began (22 January 2020), the number of infected patients is slowly rising. Unfortunately, in the last 80 days, the number of infected patients is rapidly increasing, which means that the number of confirmed patients is growing much faster than the number of recovered patients. The obtained symbolic expression for estimation of the epidemiology curve follows the trend of the real data with smaller deviations. The most noticeable deviation from real data can be seen in the last 20 days where the estimated number of infected patients is smaller than those obtained from the real data.

### 3.4. Sensitivity Analysis

The variance-based sensitivity analysis method [60] (Sobol indices) was used to estimate the effects of each model input parameter (latitude, longitude and number of days since the outbreak began) on output parameter (number of confirmed/deceased/recovered patients for the entire U.S.). This method was used to calculate the first-order, second-order and total-effect using Python package SALib [61]. The first-order indices indicate the amount of variance in the output that can be attributed to varying each parameter individually. For the first-order indices, since there is no interaction between parameters, the sum is equal to 1. The total-effect indices represent a measure of total variance for given parameter including the interaction effects and as the result have a sum greater than 1. If the total-effect indices are substantially larger than the first-order indices, then there are likely higher-order interactions occurring. The second-order indices represent measure of variance caused by variation of two parameters. The Saltelli’s cross sampling method was utilized to perform uniform distributions for each variable which resulted in 80,000 parameters. After each estimation of confirmed/deceased/recovered cases for the entire U.S., the values obtained using sensitivity analysis are given in Table 4, Table 5 and Table 6.

As shown in Table 4, the latitude ($X_{0}$) and the number of days parameter since the outbreak began ($X_{2}$) parameter exhibit the first-order sensitivities but the longitude ($X_{1}$) parameter has very small first-order sensitivity. The total-effect indices for all three parameters are substantially higher than those values obtained for first-order indices which indicates that higher order interactions occur. The second-order indices indicate the strongest interaction between latitude ($X_{0}$) and number of days ($X_{2}$) exists, which is equal to 0.384535.

Table 5 shows that, as in the case of the number of confirmed patients, the sensitivity analysis showed that the latitude ($X_{0}$) and the number of days since the outbreak began exhibit the first-order sensitivities, while the first-order sensitivity for longitude ($X_{1}$) is minimal. Higher-order sensitivities do not occur as the total-effect values are not substantially higher than the first-order indices.

From the performed sensitivity analysis, it can be noticed that first-order indices occur for number of days parameter ($X_{2}$) while the other two are very small. However, in this case, the first-order index for longitude ($X_{1}$) is much higher than latitude ($X_{0}$). The total effect indices values are not substantially higher than the first-order values so there are no higher-order interactions occurring, as shown in Table 6.

### 3.5. Discussion

From the conducted investigation, it can be noticed that each symbolic expression obtained for each U.S. state is estimating the number of confirmed patients in each state with high accuracy. As shown in Figure 1a and Table A1, the value of achieved $R^{2}$ score is in range from 0.9406 to 0.9992. From these values, it can be concluded that all symbolic expressions are estimating the number of confirmed patients for each state with very high accuracy. To obtain the symbolic expression which was used for the estimation of the number of confirmed patients for the entire U.S., all 50 symbolic expressions for the estimation of the number of confirmed patients for each state were summed up using Equation (Equation 4). This symbolic expression achieved high accuracy of 0.9987 in the estimation of the number of confirmed patients for the entire U.S, and the comparison with the real data is shown in Figure 1b. Initially, the number of confirmed patients in the U.S. for the first 60 days since the outbreak started was negligible. However, the initial government response to inform the general public was made in mid-March of 2020, although mass gatherings in the first 60 days did occur and CDC informed the general public at the end of February of a potential local outbreak. During Days 60–250, the number of confirmed patients grew from 0 to above 7,000,000. In this period, the general public protests in almost every state for anti-lockdown measures and Black Lives Matter were the most contributing factors for increased virus spreading at that time. In the last period of Days 250–317, the number of confirmed patients in the U.S. grew extremely from 7,000,000 to 14,000,000. The most contributing factor s for virus spread at that time were presidential rallies and mass gatherings in South Dakota. The symbolic expression obtained to estimate the number of the confirmed patients for the entire U.S., which was obtained as the summation of symbolic expressions for the estimation of the number of confirmed patients for each state, follows the real data with smaller deviations in the ranges 120–170 and 300–317 days. The first interval can be divided into sub-intervals where in the first (120–150 days) the symbolic expressions are underestimated the number of confirmed patients, while in the second subinterval (150–170 days) the symbolic expression overestimated the number of confirmed patients. In the second interval (300–317 days), the symbolic expressions underestimated the number of confirmed patients.

The same procedure was adopted for the estimation of the number of deceased and recovered patients. First, the symbolic expressions for the estimation of the number of deceased and recovered patients for each state were obtained, and then all 50 equations were summed up to obtain symbolic expressions for estimation the number of deceased or recovered patients. As shown in Figure 2a and Table A2, the value of achieved $R^{2}$ score with symbolic expressions for the estimation of the number of deceased patients for each state range from 0.9404 to 0.99984. The obtained symbolic expressions for the estimation of the number of deceased patients for each state were used in Equation (Equation 5) to obtain the symbolic expression for the estimation of the number of deceased patients for the entire U.S., which achieved an accuracy of 0.9997. As shown in Figure 2b the curve of deceased patients has a similar trend as the number of confirmed patients curve. In the first 60 days since the outbreak, the number of deceased patients is extremely small. In the interval from 60 to 150 days since the outbreak began, the number of deceased patients rapidly increase to almost 125,000. This increase in the number of deceased patients can be attributed to the protests that occurred in this period, which contributed to the virus spreading as well as the high mortality rate in the age groups higher than 35 years, according to Woolf et al. [59]. After 150 days, the number of deceased patients has an almost linear trend and the number of deceased patients after 300 days since the outbreak began is near 250,000. In the interval between 300 and 317 days since the outbreak began, the number of deceased patients has again rapid growth. As in the case of the number of confirmed patients, the presidential election campaign did have some influence on spreading the virus among those who attended these gatherings. The symbolic expression obtained for the estimation of the number of deceased patients follows the trend of the real data with small oscillations in the interval between 60 and 100 days. In the interval between 300 and 317 days since the outbreak began, the symbolic expression underestimated the number of deceased patients by 3%. From this comparison and the achieved $R^{2}$ score in estimation of the number of deceased patients for the entire U.S., it can be concluded that the symbolic expression estimates the number of deceased patients with high accuracy.

The $R^{2}$ score achieved with symbolic expressions for the estimation of the number of recovered patients for each state, as shown in Figure 3a and Table A3, was in range from 0.9797 to 0.99955. The symbolic expressions for the estimation of the number of recovered patients for each state were summed up using Equation (Equation 6) to obtain the symbolic expression for the estimation of the number of recovered patients for the entire U.S., and this symbolic expression achieved an accuracy of 0.9996. As shown in Figure 3b, it can be noticed that, for the first 60 days since the outbreak began, there were very few recovered patients. After 60 days, the number of recovered patients almost exponentially increased, reaching 810,000 recovered patients after 317 days. The symbolic expression for the estimation of the number of recovered patients very accurately follows the number of recovered patients when compared to the real data. The only deviation from the real data in the case of the number of confirmed and deceased patients can be noticed at the end where the symbolic expression slightly underestimated the number of recovered patients. The number of recovered patients, in general, is high when compared to the number of deceased patients. However, this number could be higher if the government at the time the outbreak in the U.S. started enforced rigorous restrictions which were already advised by CDC and healthcare professions several times in the investigated period.

The symbolic expressions for the estimation of the number of confirmed, deceased and recovered patients are obtained were all used to obtain the symbolic expression for the estimation of epidemiology curve. From the obtained solution, which is graphically represented in Figure 5, after 30 days since the outbreak began, there is a small increase in the number of infected (actual) patients. From 30 to 130 days since the outbreak began, there is an increase in the number of infected patients to almost 110,000. In the range from 150 to 250 days since the outbreak began, there is an increase in the number of infected patients to approximately 250,000. After 250 days, the number of infected patients had a huge increase, which means that the number of confirmed patients rapidly increased with a smaller increase in the number of deceased and recovered patients. The symbolic expression for the estimation of the number of infected patients, when compared to the real data, follows the trend of infected patients with smaller oscillations in the range from 100 to 150 days, where symbolic expression slightly underestimated the number of infected patients. In the range from 150 to 180 days, the symbolic expression overestimated the number of infected patients. In the range from 180 to 300 days, there are some small oscillations in the estimation of the number of infected patients. The most noticeable difference in the estimation of the number of infected patients is noticed in the range from 300 to 317 days since the outbreak began, where symbolic expression underestimated the number of infected patients. This underestimation of the number of infected patients arises from an underestimation of symbolic expressions obtained for the estimation of the number of confirmed, deceased and recovered patients.

Based on the conducted investigation, it can be noticed that the latitude and longitude of the central location, and the number of days since the outbreak began can be used as the input variables to obtain symbolic expressions for the estimation of the number of confirmed, deceased and recovered patients for each state. However, it should be mentioned that lower accuracies achieved with symbolic expressions for the estimation of the number of confirmed, deceased and recovered patients will have some influence when they are summed up to estimate the number of confirmed, deceased and recovered patients for the entire U.S. and finally the epidemiology curve. The underestimations made by symbolic expressions for confirmed, deceased and recovered patients in the interval from 300 to 317 days are noticed when the estimated epidemiology curve is compared with the epidemiology curve obtained from the real data.

The failure of the general public and the government officials to take necessary steps to prevent viral transmission made the entire U.S. vulnerable. The lack of necessary steps allowed COVID-19 to become one of the leading causes of death for those aged 35 years or older. The development and implementation of the vaccine offer some prospects, but this solution will not come soon enough to avoid an increased number of COVID-19-related hospitalizations and death.

From the performed sensitivity analysis on symbolic expression for the estimation of the number of confirmed cases for the entire U.S., it can be concluded that latitude ($X_{0}$) and the number of days since the outbreak began ($X_{2}$) have high variances, contributing to the model variance, while the longitude ($X_{1}$) accounts for only 0.047% of the model variation. The second-order variation confirms that the latitude and the number of days since the outbreak began together have high contribution to the model variance. The sensitivity analysis performed on symbolic expressions for the estimation of the number of deceased and recovered cases for the entire U.S. showed that the latitude and the number of days since the outbreak began exhibit first-order sensitivities while the longitude first order sensitivity is much smaller when compared to the other two parameters. The higher-order sensitivities do not occur due to the fact the total-effect values are not substantially higher than the first-order indices.

## 4. Conclusions

In this study, the GP algorithm was utilized on a publicly available dataset to obtain symbolic expressions for the estimation of the number of confirmed, deceased and recovered patients for each state. The symbolic expressions for the estimation of the numbers of confirmed deceased and recovered patients were summed to obtain symbolic expressions for the estimation of the number of confirmed, recovered and deceased patients for the entire U.S. The equation for the estimation of the number of confirmed, deceased and recovered patients for the entire U.S. was used to obtain the equation for the estimation of the epidemiology curve, which estimates the real epidemiology curve with high accuracy. From the extensively conducted investigations, the following conclusions can be drawn:
The GP algorithm can be utilized to obtain symbolic expressions for each U.S. state based on the latitude and longitude of their central location and day as an input variable to estimate the number of confirmed/deceased/recovered patients for the aforementioned state.The obtained symbolic expressions for the estimation of the number of confirmed/deceased/recovered patients for each state can be summed to obtain the symbolic expression for the estimation of the number of confirmed/deceased/recovered patients for the entire U.S. with high accuracy.Symbolic expressions for the estimation of the number of confirmed, deceased and recovered patients of the entire U.S. can be combined to obtain the symbolic expression for the estimation of the epidemiology curve with very high accuracy.

## Figures and Tables

**Figure 1 ijerph-18-00959-f001:**
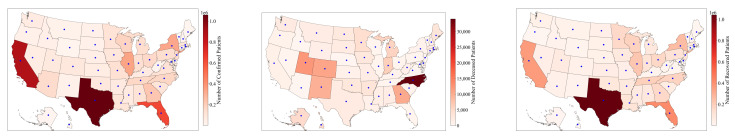
The number of confirmed, deceased and recovered patients in the U.S. on 3 December 2020.

**Figure 2 ijerph-18-00959-f002:**
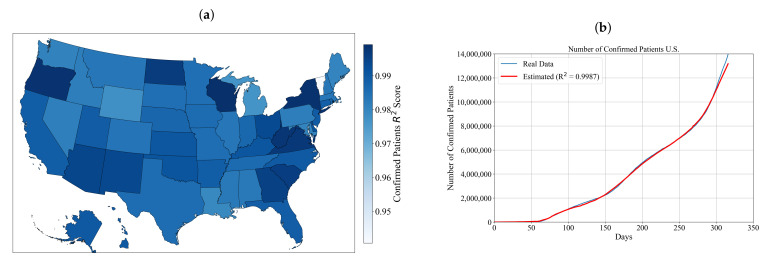
The obtained results for deceased patients: (**a**) the accuracy of obtained symbolic expressions used for estimation of number of confirmed patients for each state achieved on the testing dataset in terms of $R^{2}$ score; and (**b**) the comparison of estimated and actual numbers of confirmed patients through time.

**Figure 3 ijerph-18-00959-f003:**
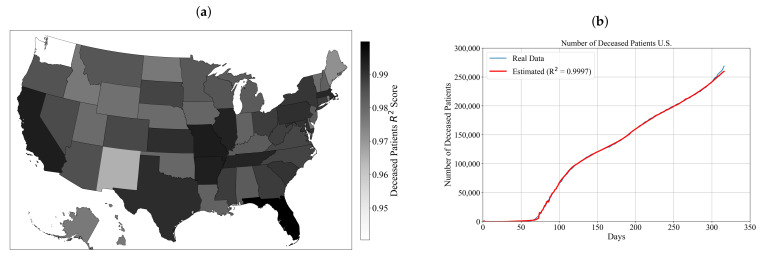
The obtained results for deceased patients: (**a**) the accuracy of obtained symbolic expressions used for estimation of number of deceased patients for each state achieved on the testing dataset in terms of $R^{2}$ score; and (**b**) the comparison of estimated and actual numbers of deceased patients through time.

**Figure 4 ijerph-18-00959-f004:**
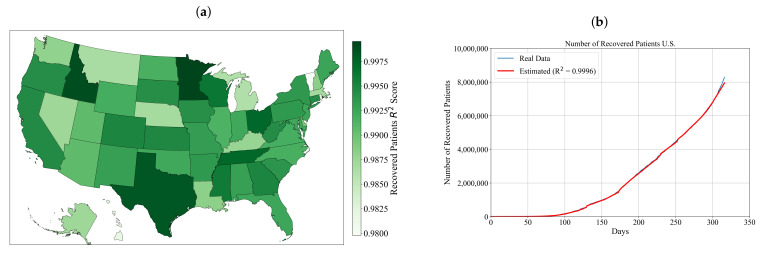
The obtained results for deceased patients: (**a**) the accuracy of obtained symbolic expressions used for estimation of number of recovered patients for each state achieved on the testing dataset in terms of $R^{2}$ score; and (**b**) the comparison of estimated and actual numbers of recovered patients through time.

**Figure 5 ijerph-18-00959-f005:**
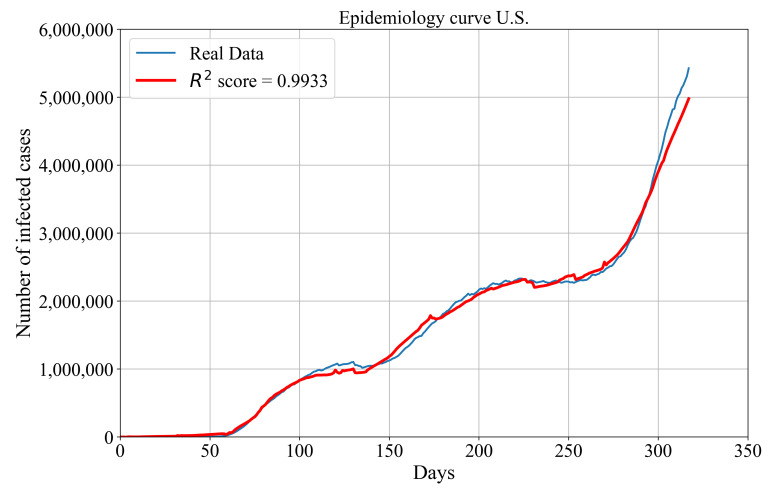
The comparison of estimated actual patients in the U.S. with the real data from the dataset.

**Table 1 ijerph-18-00959-t001:** The last ten instances of the confirmed number of patients dataset for state of Alabama.

Instance Number	Latitude	Longitude	Day	Number of Confirmed Patients
308	32.318230	−86.902298	308	236,865
309	32.318230	−86.902298	309	239,318
310	32.318230	−86.902298	310	241,957
311	32.318230	−86.902298	311	242,874
312	32.318230	−86.902298	312	244,993
313	32.318230	−86.902298	313	247,229
314	32.318230	−86.902298	314	249,524
315	32.318230	−86.902298	315	252,900
316	32.318230	−86.902298	316	256,828
317	32.318230	−86.902298	317	260,359

**Table 2 ijerph-18-00959-t002:** Input and output parameters defined and used in GP to obtain symbolic expression for estimation of number of confirmed/deceased patients for each state and recovered patients for the entire U.S.

Parameters	Confirmed Patients Analysis	Deceased Patients Analysis	Recovered Patients Analysis
Latitude	$X_{0}$	$X_{0}$	$X_{0}$
Longitude	$X_{1}$	$X_{1}$	$X_{1}$
Day	$X_{2}$	$X_{2}$	$X_{2}$
Number of patients per day	$y_{Confirmed}$	$y_{Deceased}$	$y_{Recovered}$

**Table 3 ijerph-18-00959-t003:** The list of GP parameters used to obtain symbolic expressions for the estimation of the number of confirmed patients for each state.

Parameter	Confirmed		Deceased		Recovered	
	Lower Bound	Upper Bound	Lower Bound	Upper Bound	Lower Bound	Upper Bound
Population Size	200	1000	1000	2000	1000	2000
Number of generations	100	200	100	200	100	200
Tournament Size	20	100	100	200	100	200
Tree Depth	3–6	7–12	3–6	7–12	3–6	7–12
Crossover coefficient	0.9	1	0.9	1	0.9	1
Subtree mutation coefficient	0.001	0.1	0.001	0.1	0.001	0.1
Hoist mutation coefficient	0.001	0.1	0.001	0.1	0.001	0.1
Point mutation coefficient	0.001	0.1	0.001	0.1	0.001	0.1
Stopping criteria	0.001	1	0.001	1	0.001	1
Maximum number of samples	0.9	1	0.9	1	0.9	1
Constant range	−100,000	100,000	−100,000	100,000	−100,000	100,000
Parsimony coefficient	0.1	2	0.01	0.2	0.1	2

**Table 4 ijerph-18-00959-t004:** First-order and total-effect Sobol indices measuring model sensitivity to parameters in symbolic expression for the estimation of the number of confirmed cases for the entire U.S. ($X_{0}$ latitude, $X_{1}$ longitude and $X_{2}$ day).

Variable	Distribution	Sobol Indices	
		First-Order	Total-Effect
$X_{0}$	19.74176, 66.16051	0.592586	1.036779
$X_{1}$	−155.844, −69.9722	0.00047	0.00417
$X_{2}$	(0, 317)	0.367189	1.15179916

**Table 5 ijerph-18-00959-t005:** First-order and total-effect Sobol indices measuring model sensitivity to parameters in symbolic expression for the estimation of the number of deceased cases for the entire U.S. ($X_{0}$ latitude, $X_{1}$ longitude and $X_{2}$ day).

Variable	Distribution	Sobol Indices	
		First-Order	Total-Effect
$X_{0}$	19.74176, 66.16051	0.10282	0.124253
$X_{1}$	−155.844, −69.9722	0.002036	0.0127
$X_{2}$	(0, 317)	0.87679	0.99746

**Table 6 ijerph-18-00959-t006:** First-order and total-effect Sobol indices measuring model sensitivity to parameters in symbolic expression for the estimation of the number of recovered cases for the entire U.S.

Variable	Distribution	Sobol Indices	
		First-Order	Total-Effect
$X_{0}$	19.74176, 66.16051	0.13976	0.010748
$X_{1}$	−155.844, −69.9722	0.016542	0.13089
$X_{2}$	(0, 317)	0.836388	0.986203

## Data Availability

The obtained hyperparameters and the transformed dataset used in the research is made available at https://github.com/RitehAIandRobot/CovidUSA-RegressionData.

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
