# Peer review of "Estimation of COVID-19 Epidemiology Curve of the United States Using Genetic Programming Algorithm"

_ijerph, 2021, doi:10.3390/ijerph18030959_

Round 1

Reviewer 1 Report

The authors aim to obtain a symbolic expression for the estimation of the number of confirmed, deceased and recovered patients for the states of the US using a GP algorithm. The authors motivate their work properly, and include a thorough literature review. The authors use publicly available data for their task, obtained from the Johns Hopkins dashboard. Apart from covid data, the use the latitudes and longitudes of the states and the calendar day. The obtained syntax trees are somewhat complex, but they achieved good accuracy on the test dataset. Some of the design choices of their approach need further clarification however, and the discussion of their results is lacking in depth.

The authors used the dashboard provided by Johns Hopkins university as a data source for covid cases, but their model only spans a single country, the USA. They state that they can’t model DC, Puerto Rico, Virgin Islands, etc. because lack of data on the Johns Hopkins website. If they are modelling the U.S., why not use data provided by CDC instead? That includes data for the territories missing from the Johns Hopkins website. (https://covid.cdc.gov/covid-data-tracker)

My main concern with this paper is the choice of input attributes. The authors consider a single well-developed country in their work. This allows them to have access to plenty of variables that can be related to epidemic spreading: population density, climate, travel, etc. Yet they only include longitude and latitude (apart from elapsed time) as their input. Why do the authors think they are so relevant to disease spreading? What is their expected role in covid spreading? What is their relevance in covid spreading according to the output of their proposed model? These are critical questions that need detailed answers in the paper and have none. Simply stating that the model has high accuracy is not enough.

In what specific way did the authors separate their input into training and test datasets?

I’m not sure what the input variable X2 is. The data section simply says the day since the infection started. On page 10 line 232 it reads: “X2 the number of confirmed patients for each day since the COVID-19 outbreak started in each federal state.” But isn’t that the output of the model?

The discussion is very repetitive and does not have enough focus on the output of the model (the relationship between the input variables and the output), while the hyperparameters are given way too much detail. The Tables 3 to 5 take up too much space, I suggest moving them to supplementary. For the same reason, similar figures could be moved into panels.

Minor comments

The description of the GP algorithm and the syntax trees is a but tedious. It can be assumed that the reader is roughly familiar with these concepts.

This is just an optional suggestion, but maybe try to put Figures 1 to 3 into a single panel? They would take up much less space that way.

The English is readable but there are plenty of typos and grammatical mistakes that need fixing.

Author Response

Respected Reviewer,

please find attached the response to the your comments.

Thank you for your review of our manuscript,
Authors

Reviewer 2 Report

Manuscript ID: ijerph-1056792

Title: Estimation of covid-19 epidemiology curve of the united states using genetic programming algorithm  

The current work addresses the develop of epidemiological models for COVID-19 in united states. The models are constructed using genetic programming algorithms. The following remarks are provided:

  1. The authors must be more explicit when indicate numerous researchers have utilized tools included into artificial intelligence (AI) to address the mathematical modeling for COVID-19 instances. AI is a branch of computer science, which include the soft computing or computational intelligence (CI). CI can be divided into probabilistic reasoning, and functional approximation and randomized search. The first group, in turn, can be divided into probabilistic models and fuzzy logic. The second group, in turn, can be divided into evolutionary computing, swarm optimization, and machine learning [1–3]. Usually, these three latter approaches and their hybridizes (hybrid systems or learnheuristics) are the tools implemented to develop epidemiological models for COVID-19.
  2. Usually, real dataset exhibits duplicate values, outliers, error, special characters, among others. The authors could explain the procedure used dataset cleaning.
  3. The literature [4,5] indicates that some longitude (X0) and latitudes (X1) ranges influence on higher COVID-19 death rates. The authors must consider this information in the current work to justify the include X0 and X1 in the developed symbolic expressions. The following figure shows the impact of input variables uncertainty on Idaho deaths (global sensitivity analysis, reviewer). In other words, this figure allows to confirm the works cited.
  4. Available literature shows works that address the epidemiological model for COVID-19, for example, [4] indicate that higher COVID-19 death rates are observed in the [25/65] latitude and in the [-35/-125] longitude ranges, which can be verified via following figures. These latter figures were confectioned using the symbolic expression of Idaho deaths. In other words, the current work lacks main objective, i.e., it is not enough to develop models and to report that these exhibit a good fit, the authors should indicate how they will use the developed models. For example, uncertainty quantification, classification by latitude/longitude, 2D or 3D graphics, among others.
  1. Symbolic regression genetic programming exhibit two problems: bloat and overtraining. The first is defined as the increase in mean program size without a corresponding improvement in fitness. The second occurs when a developed model begins to “memorize” training data rather than “learning” to generalize from a trend [6,7]. It is generally assumed that bloat and overfitting are related. However, it has been reported that overfitting can occur in absence of bloat, and vice versa [8]. In current work, the authors avoid the bloat penalizing the parsimony coefficient. How was addressed the overfitting?
  2. The related literature indicates that genetic programming exhibits a computationally intensive nature, which makes it difficult to apply to real world problems [9,10]. The authors could explain the execution time utilized by developed codes during supervised training?
  3. The authors randomly selected the parameters of algorithms, however the related literature indicates that a chaotic selection provide better results both solution quality and execution time. Maybe a good idea will be executing the codes using logistic map, Arnold’ cat map (Figure), among others, when the symbolic model exhibits a R2<0.95, this latter value is usually used by many authors as target accuracy.
  4. The related literature indicates that metamodels must be subjected to verification, validation, and uncertainty quantification (UQ) to guarantee their robustness. Under the current context, only UQ must be considered to provide a general overview of the effect of the uncertainties in the metamodel responses [11]. In other words, a Radj close to 1 does not guarantee a good metamodel. The authors could show the effect of uncertain on developed models at least for one state of USA?

Author Response

(The authors gave the same response as above.)

Reviewer 3 Report

In this paper entitled “Estimation of COVID-19 Epidemiology Curve of the United States Using Genetic Programming Algorithm”, the authors addressed the current context of the COVID-19 outbreak and estimated the epidemiology curve for the U.S. by using genetic programming. The authors efforts should be encouraged. This paper is informative and useful, and the paper holds high potential for publication. However, to meet the increasingly high-quality standard of the journal a careful revision is needed.

Major Revision:

  1. The Paper needs some language corrections throughout the text before being published.

        Just to mention few:

        In abstract, line number 2 change “So far, there were were some                    implementations” to “So far, there were some implementations”.

        Line number 85“heart vesel is reduced by” to “heart vessel is reduced by”

  1. In introduction authors should mention what the global trend for COVID-19 has been like.
  2. Authors should also emphasize on the reason why they only focused on U.S.
  3. Authors should refer the table as “Table” instead of “Tab” throughout the paper
  4. Reframe the paragraph from line 225-227 under section “Results and Discussion”.
  5. There are number of references are not cited properly for example reference number 3,4,6,7,8. Please make sure to cite the reference in a way which helps the readers to refer to them.

Author Response

(The authors gave the same response as above.)

Reviewer 4 Report

Anđelić et al, uses GP to optimize expression trees to fit COVID data.

The article is interesting; however, the writing could be much improved. I understand that, for non-native speakers (including myself) the English usually punish the work, but my suggesting is to use some expert help to improve the writing of this article.

Some typos:

2: were were

150: can issues in the model interpretation.

187: half-anf-half

Another issue with the article is the page number. A lot of the details could be sent to supplementary material sections (i.e., tables keep only the data relevant for the discussion), details regarding the mechanism of GP. Also, the plots could be merged to a single figure with 3 subplots.

Regarding the results:

Table 2 shows the parameters used for each of the three datasets, the text did not mention why different values for each of the datasets, was the values chosen by trial and error?

The results only shown how good is the GP method to fit the data, however, the expressions were built clearly shows how complex the GP tends to go to max fit the data (i.e., getting max fitness score), if limited to simpler expressions and smaller tree density would not the resulting expression be more interpretable instead? How would that punish the fitness score? In my view point the authors were too focused getting high fitness score which could generate poor predictors.

Is the data and methods available to the readers? The authors did mention the source of the data, however, kindly provide the exact data used would allow better reproducibility of the research.

Author Response

(The authors gave the same response as above.)

Round 2

Reviewer 1 Report

The authors revised their manuscript, and most of my questions were addressed.

Please remove the following sentence, as it is inaccurate/untrue (line 186): “The other reason why the aforementioned variables were not included is that the process of collecting them would be time-consuming and the quality of quantification for certain variables, such as migration or climate, is questionable.” Collecting single country data from a developed country takes a minimal amount of time, while migration and climate factors have been studied in depth in the epidemiology literature before and their use is not questionable at all.

With the above modification the manuscript becomes adequate for publication.

Author Response

We would like to thank the First Reviewer for the revision of our manuscript.

The sentence in question has been removed from the manuscript.

Thank you again,
Authors

Reviewer 2 Report

How the authors guarantee the continuity of expressions developed? 

Author Response

We would like to thank the Second Eeviewer for the review of our manuscript. The answer to the posed question follows below, and has been included in the manuscript:

The continuity of the generated equations is guaranteed by the fact the GP uses modified versions of certain mathemathical operations. For example, in case of a square root $\sqrt{x}$, the operation is implemented as a square root of an absolute value ($\sqrt{|x|}$), to avoid calculating square roots of negative values. Another common discontinuity is division by zero. In case of division by zero, or near-zero values a protected division is used, which returns the value of 1.0. These modified operations vary between GP implementations and must be taken into account when the generated models are implemented.

Thank you again,
Authors